# TIGIT blockade improves anti-*Mycobacterium tuberculosis* immunity

**Jingyu Zhou**[1☯], **Qingluan Yang**[1☯], **Haoxin Xu**[1☯], **Huaxin Chen**[2☯], **Ning Jiang**[1,3,4], **Qinfang Ou**[2], **Mengqing Qian**[1], **Xing Lin**[1], **Yixuan Yang**[1], **Feiran Zhou**[1], **Yuzhen Xu**[1], **Qianqian Liu**[1], **Yuanyuan Liu**[1], **Yan Gao**[1], **Wenhong Zhang**[1,4], **Lingyun Shao**[1]*, **Qiaoling Ruan**[1]*

1 Department of Infectious Diseases, Shanghai Key Laboratory of Infectious Diseases and Biosafety Emergency Response, National Medical Center for Infectious Diseases, Huashan Hospital, Fudan University, Shanghai, China, 2 Department of Pulmonary Diseases, Wuxi Fifth People's Hospital, Wuxi, China, 3 Department of Biostatistics and Computational Biology, SKLG, School of Life Sciences, Fudan University, Shanghai, China, 4 Shanghai Sci-Tech Inno Center for Infection & Immunity, Shanghai, China

☯ These authors contributed equally to this work.
\* lingyun26@fudan.edu.cn; (LS) qlruan07@fudan.edu.cn (QR)

## Abstract

Despite the therapeutic benefit of immune checkpoint blockade in cancers, there is no consensus on its effect in infectious diseases. Here we investigated whether blocking the immune checkpoint T cell immunoreceptor with immunoglobulin and immunoreceptor tyrosine-based inhibitory motif domains (TIGIT) increases T cell immunity in active *Mycobacterium tuberculosis* infection. TIGIT expression in both peripheral blood and lung lesions in tuberculosis patients was assessed, and the correlation with clinical features analyzed. The functional status of TIGIT+ and TIGIT−CD8+ T cell subsets in tuberculosis patients was analyzed by flow cytometry and transcriptome analysis. To investigate the regulatory effect of TIGIT, the function of CD8+ T cells in tuberculosis patients and bacterial load in a tuberculosis mouse model were assessed after *in vitro* and *in vivo* TIGIT blockade. In active tuberculosis patients, TIGIT expression on CD8+ T cells in the peripheral blood was significantly upregulated and positively correlated with disease severity. TIGIT expression in lung lesions was significantly higher in patients with pulmonary tuberculosis than in patients infected with other pathogens. TIGIT+CD8+ T cells exhibited higher activation and differentiation levels, increased expression levels of cytokines and cytotoxic molecules, and showed gene expression features of natural killer-like cytotoxic effector CD8+ T cells. TIGIT blockade increased the ability of human CD8+ T cells to produce effector molecules and kill intracellular bacteria *in vitro*. Importantly, blocking TIGIT reduced lung bacterial burden in mice infected with *M. tuberculosis*. The findings reveal that in active tuberculosis patients, activated CD8+ T cells express TIGIT and blocking TIGIT enhances CD8+ T cell function and promotes clearance of *M.*

**Data availability statement:** All relevant data are within the manuscript and its Supporting information files.

**Funding:** This work is supported by the National Natural Science Foundation of China [82271794, to Q.R; 82302533, to Q.Y; 82101852, to Y.L]. The funders had no role in study design, data collection and analysis, decision to publish, or preparation of the manuscript.

**Competing interests:** The authors have declared that no competing interests exist.

*tuberculosis*. The findings also suggest that TIGIT limits T cell immunity in tuberculosis and implicate TIGIT blockade as a novel strategy for tuberculosis therapy.

## Author summary

Tuberculosis remains a leading public health threat, with high morbidity and mortality worldwide. It develops as *Mycobacterium tuberculosis* manages to dodge elimination by host T-cell immune response and persist as a chronic infection. Immune checkpoint TIGIT has been reported as a negative regulator of T-cell effector functions. However, its role in T-cell-mediated host defense against tuberculosis remains unclear. Our data showed that in active tuberculosis patients, TIGIT expression on peripheral CD8+ T cells upregulated significantly and correlated with more severe disease. TIGIT expression in lung lesions in patients with pulmonary tuberculosis is also higher. TIGIT+CD8+ T cells exhibited higher activation and differentiation levels, increased expression levels of effector molecules, and shared gene expression features with natural killer-like cytotoxic effector CD8+ T cells. TIGIT blockade enhanced the effector function of human CD8+ T cells, and most importantly, reduced lung bacterial burden in mice infected with *M. tuberculosis*. The findings reveal that in active tuberculosis patients activated CD8+ T cells express TIGIT and blocking TIGIT enhances CD8+ T cell function and promotes clearance of *M. tuberculosis*. The findings also suggest that TIGIT limits T cell immunity in tuberculosis and implicate TIGIT blockade as a novel strategy for tuberculosis therapy.

## Introduction

Tuberculosis caused by *Mycobacterium tuberculosis* (MTB) infection remains a leading public health threat, with high morbidity and mortality worldwide [1]. Approximately 25% of the global population is infected with MTB, among which 5%–15% will progress to active tuberculosis (ATB) during their lifetime [2]. In the case of latent tuberculosis infection (LTBI), chronic MTB antigen stimulation may persist, resulting in compromised T cell effector function and succumb to ATB [3]. T cell immunity has a deterministic effect on the outcome of MTB infection [4]. The importance of CD4+ T cells in the control of MTB has been proven by substantial evidence. Despite past controversy, mounting evidence now support an indispensable role of CD8+ T cells are important in managing MTB infection [5,6]. Recent data from non-human primate models further unveiled a crucial role of multifunctional CD8+ T cells in robust protection against MTB infection or vaccination [7–9]. In human, the control of MTB has been linked to generation of multifunctional CD8+ T cell, which can kill intracellular mycobacteria not only via cytotoxic activity and release of antimicrobial peptide granulysin, but also via production of cytokines and subsequent activation of various immune cells [5,6]. Nevertheless, there are still gaps in understanding the mechanism of action of CD8+ T cells and the regulation of their functions.

During chronic infection and cancer, high antigenic loads continually stimulate T cells and leading to a step-wise loss of effector functions [10]. Dysfunctional T cells are characterized by the upregulation of multiple immune checkpoints, such as programmed cell death protein 1 (PD-1), T cell immunoglobulin and mucin domain-3 (TIM-3) [10]. Although immune checkpoints are transiently expressed in functional effector T cells during activation, higher and sustained expression of immune checkpoints is a hallmark of exhausted T cells [10]. Blockade of these checkpoint receptors could result in therapeutic benefit by reversing the dysfunctional state of anti-tumor T-cell immunity [11]. However, the effect of checkpoint inhibition in infectious diseases remains inconclusive [12,13]. Given the prevalence of checkpoint blockade-associated tuberculosis in cancer patients [14], the role of immune checkpoints in infections must be clarified.

T cell immunoreceptor with immunoglobulin and immunoreceptor tyrosine-based inhibitory motif domains (TIGIT) is an immune checkpoint that was recently highlighted as a candidate target in cancer immunotherapy [15]. Upregulated TIGIT expression on T cells is reportedly associated with impaired immune response in several viral and parasitic infections [16–20]. TIGIT blockade alone or combined blockade with PD-1 can partially restored T cell function against infection [16,19,21]. However, how TIGIT participates in the pathogenesis of tuberculosis is unclear.

We investigated the role of TIGIT in T cell immunity in the setting of human MTB infection. Accordingly, we compared the T cell expression of TIGIT among individuals with either ATB or LTBI, and investigated the phenotypic and transcriptional profiles of TIGIT-expressing CD8$^+$ T cells. Finally, the effect of TIGIT blockade on the T cell in both ATB patients and MTB infected mice was assessed.

## Results

### Expression of TIGIT on CD8$^+$ T cells is upregulated in individuals with ATB and is correlated with disease severity

We first assessed in blood T cell subsets of individuals with ATB, LTBI, and healthy controls (Fig 1A). The expression of TIGIT on CD8$^+$ T cells, but not CD4$^+$ T cells, elevated significantly in individuals with ATB as compared to individuals with LTBI and healthy controls without MTB infection (Fig 1A). TIGIT-expressing cells comprised > 60% of effector memory (T$_{em}$) and effector memory-expressing CD45RA (T$_{emra}$) CD8$^+$ T cell subsets in ATB individuals. Both the proportions were significantly higher than those in healthy controls (Fig 1B). Increased expression of the CD155 TIGIT ligand on CD14$^+$ monocytes was observed in individuals with ATB compared to healthy controls (Fig 1C). We subsequently compared TIGIT expression in blood samples from individuals with ATB with different clinical symptoms. TIGIT expression on blood CD8$^+$ T cells in ATB patients with bilateral lung lesions was higher than in those with unilateral lesion (Fig 1D). The frequencies of TIGIT$^+$CD4$^+$ T cells was comparable for ATB patients (S1 Fig). We therefore determined the expression of TIGIT in lung lesion sections (Fig 1E). Due to the difficulty of obtaining lung sections from healthy and LTBI individuals, individuals infected with other pathogens were included as controls. The resulting histochemistry score showed a more than 10-fold higher lung TIGIT expression level in individuals with ATB (Fig 1E). Together, these results revealed an upregulated expression of TIGIT in both peripheral blood and lung lesions during active tuberculosis. Furthermore, TIGIT expression was associated with disease severity.

### Expression of TIGIT on activated effector CD8$^+$ T cells in individuals with ATB

We next assessed the effector phenotype and functional status of TIGIT$^+$ and TIGIT$^-$ CD8$^+$ T cell populations. TIGIT$^+$CD8$^+$ T cells displayed higher frequencies of the CD25 and CD69 activation markers (Fig 2A), and comprised significantly higher percentages of CCR7$^-$CD45RA$^+$ T$_{emra}$, CCR7$^-$CR45RA$^-$ T$_{em}$, and CCR7$^+$CD45RA$^-$ T$_{cm}$ cells, as compared to the TIGIT$^-$ subset (Fig 2B). TIGIT is considered a marker of immune exhaustion in multiple cancers [22]. To determine whether TIGIT$^+$CD8$^+$ T cells retain features of immune exhaustion, we stained blood cells from ATB individuals for PD-1 and TIM-3. With most of the CD8$^+$ T cells being TIGIT single-positive, we observed significantly more PD-1-expressing cells within the TIGIT$^+$CD8$^+$ T cell compartment, comprising > 16% of PD-1$^+$TIM-3$^-$ cells and < 1% PD-1$^+$TIM-3$^+$ cells (Fig 2C).

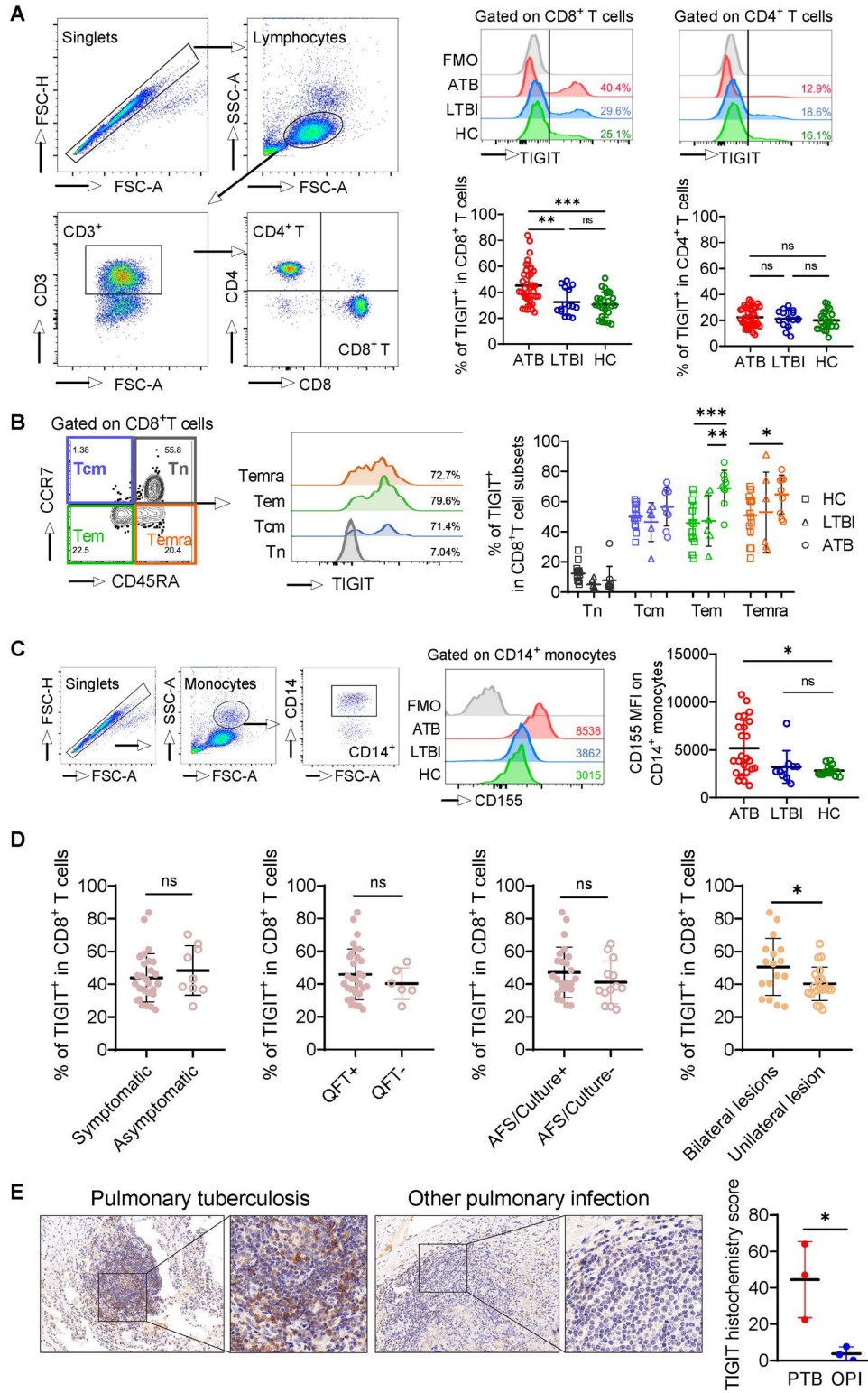

**Fig 1. Expression of TIGIT on CD8 +T cells is upregulated in individuals with ATB and correlates with disease severity.** (A) Gating strategy, representative histograms of TIGIT expression, and the frequencies of TIGIT-expressing cells among CD8+ and CD4+ T cell populations from blood in individuals with ATB (n = 37), LTBI (n = 15), and healthy control (HC, n = 25). (B) Representative flow contour plot of blood CD8+ T cell populations expressing CCR7 and/or CD45RA, and histograms of TIGIT expression on CCR7−CD45RA+ (T$_{emra}$), CCR7−CD45RA− (T$_{em}$), CCR7+CD45RA− (T$_{cm}$), and

CCR7+CD45RA+ (T_n) CD8+ T cell subsets in an ATB patient. Summarized frequencies of TIGIT-expressing cells in T_emra, T_em, T_cm, and T_n CD8+ T cell subsets from blood in individuals with ATB (n=9), LTBI (n=6), and HC (n=14). (C) Gating strategy, representative histograms of CD155 expression, and the cumulative level of CD155 MFI among CD14+ monocyte populations from blood in individuals with ATB (n=24), and LTBI (n=10), as well as HCs (n=15). (D) Frequencies of TIGIT-expressing cells on blood CD8+ T cells among patients with ATB (n=37) with different symptoms, interferon-gamma release assay, MTB test results, and chest computed tomography scan results. Symptomatic refers to cough over 2 weeks, chest pain, fatigue, weight loss, fever, night sweats, dyspnea, and/or hemoptysis. (E) Expression of TIGIT in lung lesion sections of representative individuals with ATB and another pulmonary infection. Summarized histochemistry score calculated by multiplying the quantity and intensity scores for semi-quantification of TIGIT staining in lung lesions in individuals with ATB (PTB, n=3) and another pulmonary infection (OPI, n=3). Data are presented as mean±standard deviation in panels A, D, and E, and as the median with IQR in panels B and C. Statistical significance ($P<0.05$) was obtained using a one-way ANOVA with Bonferroni post-test (Kruskal-Wallis with Dunn post-test) (A, B) or Kruskal-Wallis (C) or unpaired t (Mann-Whitney U) test (D, E). *, $P<0.05$; **, $P<0.01$; ***, $P<0.001$; ns, not statistically significant.

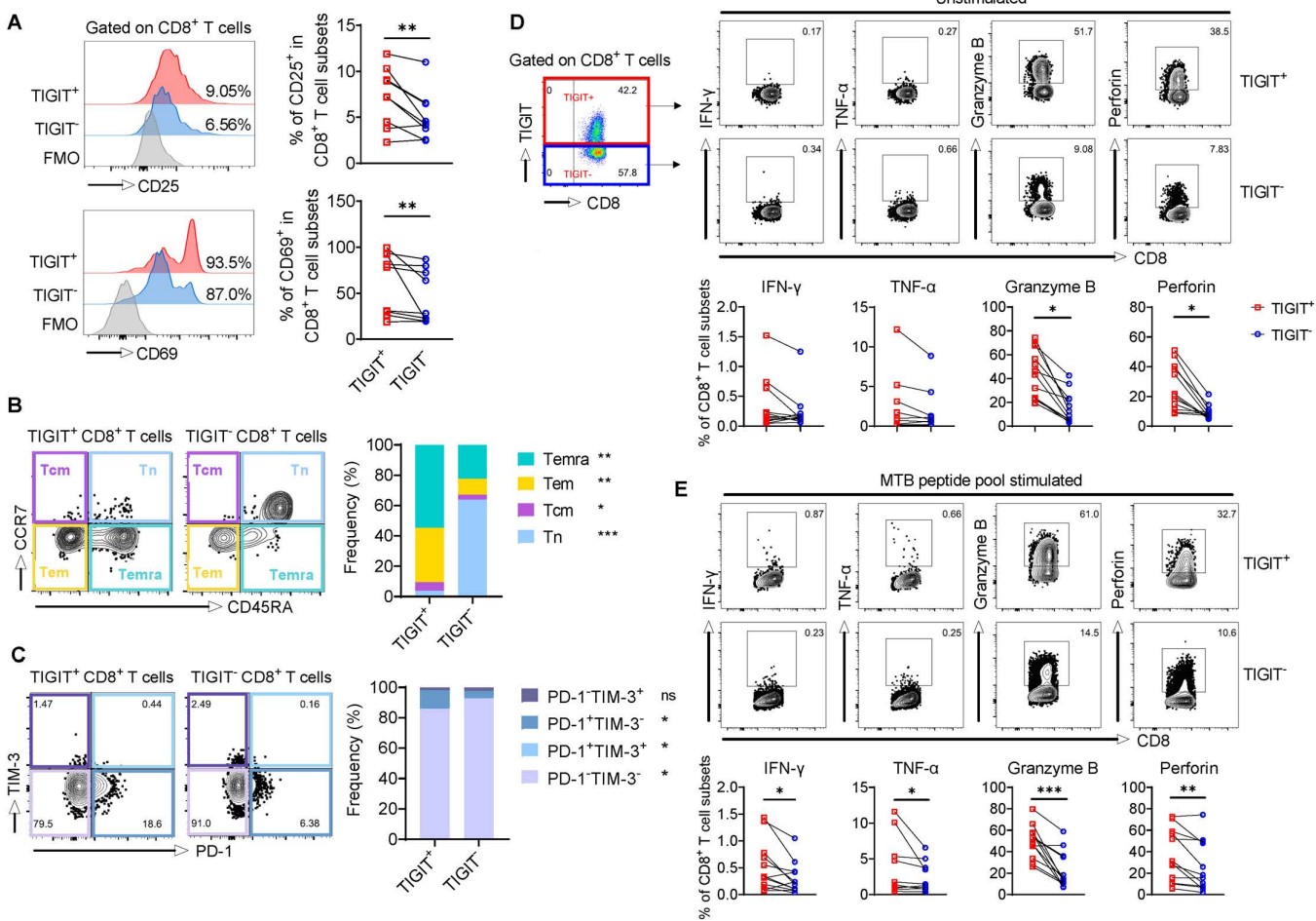

**Fig 2. TIGIT expression on activated effector CD8 +T cell populations in individuals with ATB.** (A) Representative histograms of CD25 and CD69 expression, and cumulative frequencies of CD25+ and CD69+ cells among CD8+ T cell populations from blood in individuals with ATB (n=9). (B) Representative flow contour plots of CCR7 and CD45RA expression and cumulative composition proportion of effector/memory subsets among TIGIT+ and TIGIT−CD8+ T cell populations in individuals with ATB (n=9). (C) Representative flow contour plots of TIM-3 and PD-1 expression and cumulative composition proportion of PD-1−TIM-3+, PD-1+TIM-3−, PD-1+TIM-3+, and PD-1−TIM-3− subsets among TIGIT+ and TIGIT−CD8+ T cell populations in individuals with ATB (n=9). Gating strategy, representative flow contour plots and summarized frequencies of IFN-γ, TNF-α, granzyme B, and perforin expression in unstimulated (D) and MTB peptide stimulated (E) TIGIT+ and TIGIT−CD8+ T cell populations from individuals with ATB (n=12). Data are presented as individual values in panels A, D, and E, and mean±standard deviation in panels B and C. Statistical significance ($P<0.05$) was obtained using a paired t (Wilcoxon signed rank) test or one-way ANOVA with Bonferroni post-test. *, $P<0.05$; **, $P<0.01$; ***, $P<0.001$; ns, not statistically significant.

Given the regulatory role of TIGIT in previous studies, we assessed the function of CD8$^+$ T cells from ATB individuals to determine whether the TIGIT$^+$CD8$^+$ T cells show sign of exhaustion. Intracellular cytokine staining of IFN-γ, TNF-α, granzyme B, and perforin revealed that resting state TIGIT$^+$CD8$^+$ T cells produced significantly more cytotoxic effector molecules granzyme B and perforin (Fig 2D). In response to MTB peptide stimulation, the TIGIT$^+$ compartment produced higher amounts of all four kinds of effector molecules compared to TIGIT$^-$ cells (Fig 2E). Analysis of TIGIT expression on the molecule-expressing MTB-responsive CD8$^+$ T cells showed a significantly higher level of TIGIT expression on either IFN-γ- or granzyme B-producing CD8$^+$ T-cell subset, among which the IFN-γ$^+$ granzyme B$^+$ multifunctional subset had the highest TIGIT expression (S2 Fig). Moreover, we observed a slightly higher expression of the Ki-67 cellular proliferation marker in the TIGIT$^+$ compartment (S3 Fig). The collective data revealed the higher frequency of distribution of activated effector cells that responded well to MTB in the TIGIT$^+$CD8$^+$ T cell subset, while few TIGIT$^+$CD8$^+$ T cells exhibited typical features of exhaustion.

## TIGIT$^+$CD8$^+$ T cells from individuals with ATB share transcriptional features with natural killer (NK)-like cytotoxic effector CD8$^+$ T cells

We then performed bulk RNA-sequencing to assessed the transcriptional profile of TIGIT$^+$ CD8$^+$ T cells. As co-expression of PD-1 and TIGIT was observed on CD8$^+$ T cells, RNA-seq was performed using four populations of CD8$^+$ T cells sorted from blood of four patients with ATB based on the expression of TIGIT and PD-1, to eliminate the probable regulatory effect of PD-1 on TIGIT$^+$ CD8$^+$ T cells. Distinct gene expression profiles of the TIGIT$^+$ subsets were evident (Fig 3A). The 375 overlapping TIGIT-related DEGs were enriched in secretory granules, T cell activation, proliferation, and T cell mediated cytotoxicity (Fig 3B), and pathways related to cytokine-cytokine receptor interaction, NF-κB and PI3K-Akt signaling (Fig 3B).

Moreover, multiple genes encoding NK receptor and cytotoxic granules associated with NK-like cytotoxicity and anti-tuberculosis effector function were more highly expressed in TIGIT$^+$PD-1$^-$ cells (Fig 3C). TIGIT$^+$CD8$^+$ T cell subsets also expressed higher levels of chemokine and chemokine receptor genes associated with CD8$^+$ T cell differentiation, such as *XCL2*, but lower levels of *CCR2* and multiple cytokine receptor genes associated with T cell homeostasis and homing, such as *IL23R* and *IL7R* (Fig 3C). Transcription factor T-bet and eomesodermin, which are key regulators of CD8$^+$ T cell differentiation and development of effector function [23], were also upregulated in the TIGIT$^+$PD-1$^-$ subsets. These upregulations were subsequently confirmed by flow cytometry (Fig 3C and 3D). Such a transcriptional signature indicated that TIGIT$^+$CD8$^+$ T cells possess the capacity to produce anti-tuberculosis effector molecules and phenotypic features similar to NK cells, instead of being functionally exhausted, in individuals with ATB.

## Blocking TIGIT promotes clearance of mycobacteria by CD8$^+$ T cells *in vitro*

To determine how TIGIT affect host anti-tuberculosis immune response, we next assessed the effect of *in vitro* TIGIT blockade on the functions of CD8$^+$ T cells from individuals with ATB. We observed a higher production of IFN-γ and granzyme B by CD8$^+$ T cells in response to MTB peptide stimulation after TIGIT blockade (Fig 4A). Cumulative data of the frequency of carboxyfluorescein succinimidyl ester (CFSE)$^{low}$ cells among CD8$^+$ T cells after a 5-day culture revealed that TIGIT blockade significantly improved T cell receptor-mediated CD8$^+$ T cell proliferation (Fig 4B). To further clarify the effect of TIGIT on T cell mediated control of MTB infection, Bacillus Calmette-Guérin (BCG)-infected monocyte-derived macrophages (MDM) were co-cultured with autologous sorted CD8$^+$ T cells from the blood of individuals with ATB. TIGIT blockade led to a significant reduction in BCG colony forming units (CFU) counts in co-culture cell lysates by an average of $0.74 \times 10^4$/ml, as compared to the isotype control antibody treated group (Fig 4C). These results supported an inhibitory effect of TIGIT on CD8$^+$ T cell-mediated anti-tuberculosis immune response, as *in vitro* TIGIT blockade enhanced effector molecule production, proliferation, and killing of intracellular infected BCG.

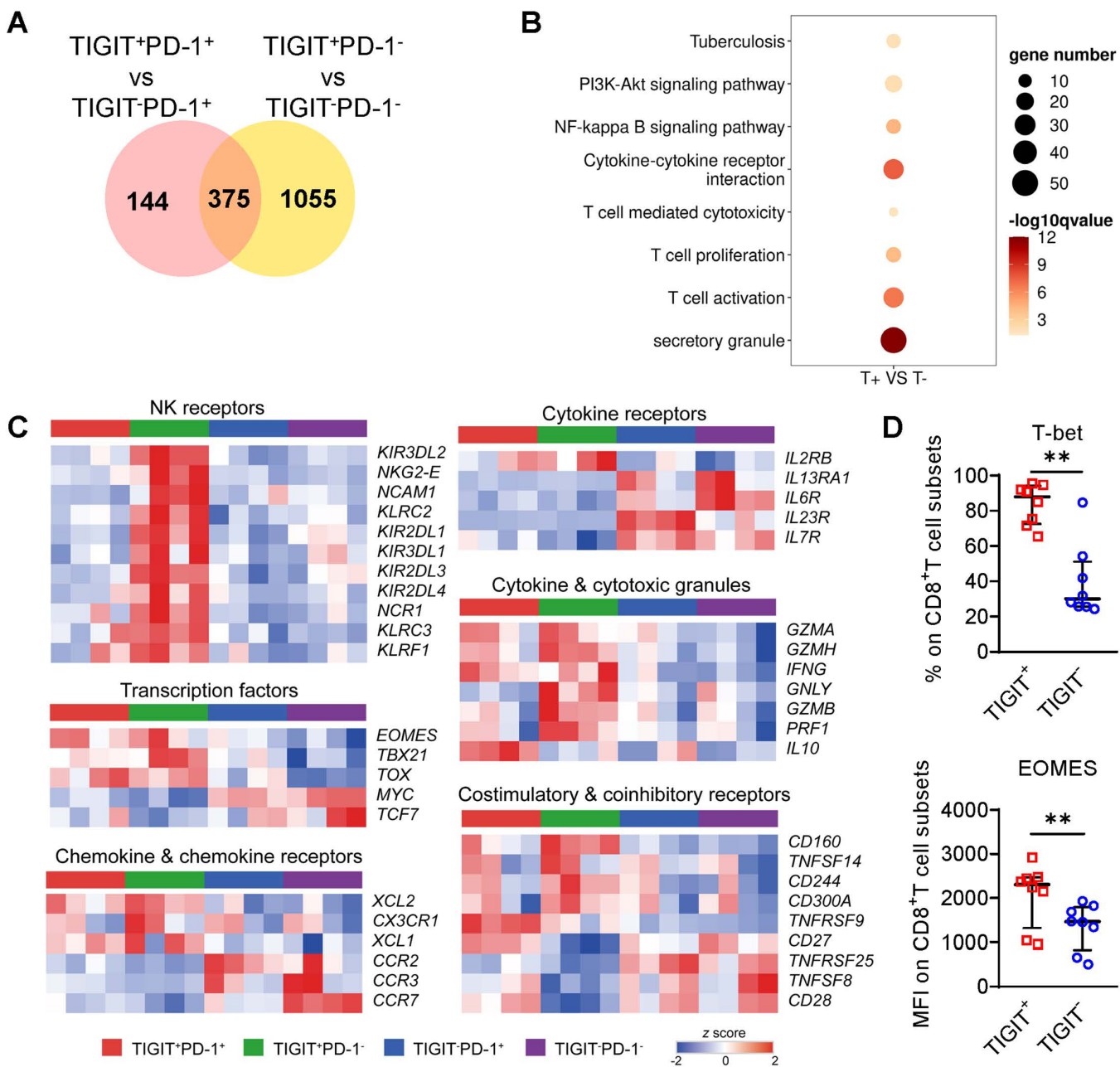

**Fig 3. TIGIT⁺CD8⁺ T cells from ATB individuals share transcriptional features with NK-like cytotoxic effector CD8⁺ T cells.** Bulk RNA-seq and gene expression analysis were performed using sorted TIGIT⁺PD-1⁺, TIGIT⁺PD-1⁻, TIGIT⁻PD-1⁺, and TIGIT⁻PD-1⁻ blood CD8⁺ T cell subsets from four individuals with ATB. (A) Venn diagram of differentially expressed gene (DEGs) related to TIGIT expression in CD8⁺ T cell subsets. (B) Kyoto Encyclopedia of Genes and Genomes pathway (upper four) and Gene Ontology functional (lower four) enrichment analysis of 375 overlapping DEGs in TIGIT⁺CD8⁺ T cell subsets compared to the TIGIT⁻ subsets. (C) Heatmaps showing scaled (z score) gene expression of representative gene sets encoding NK receptors, transcription factors, chemokine and chemokine receptors, cytokine receptors, cytokine and cytotoxic granules, and costimulatory and coinhibitory receptors in TIGIT⁺ and TIGIT⁻CD8⁺ T cell subsets (upregulated in red, down-regulated in blue). (D) Cumulative expression of transcription factors T-bet and eomesodermin (EOMES) in TIGIT⁺ and TIGIT⁻CD8⁺T cell subsets from blood of patients with ATB (n=8). Data are presented as median with IQR in (D). Statistical significance ($P < 0.05$) was obtained using a Wilcoxon signed rank test. **, $P < 0.01$.

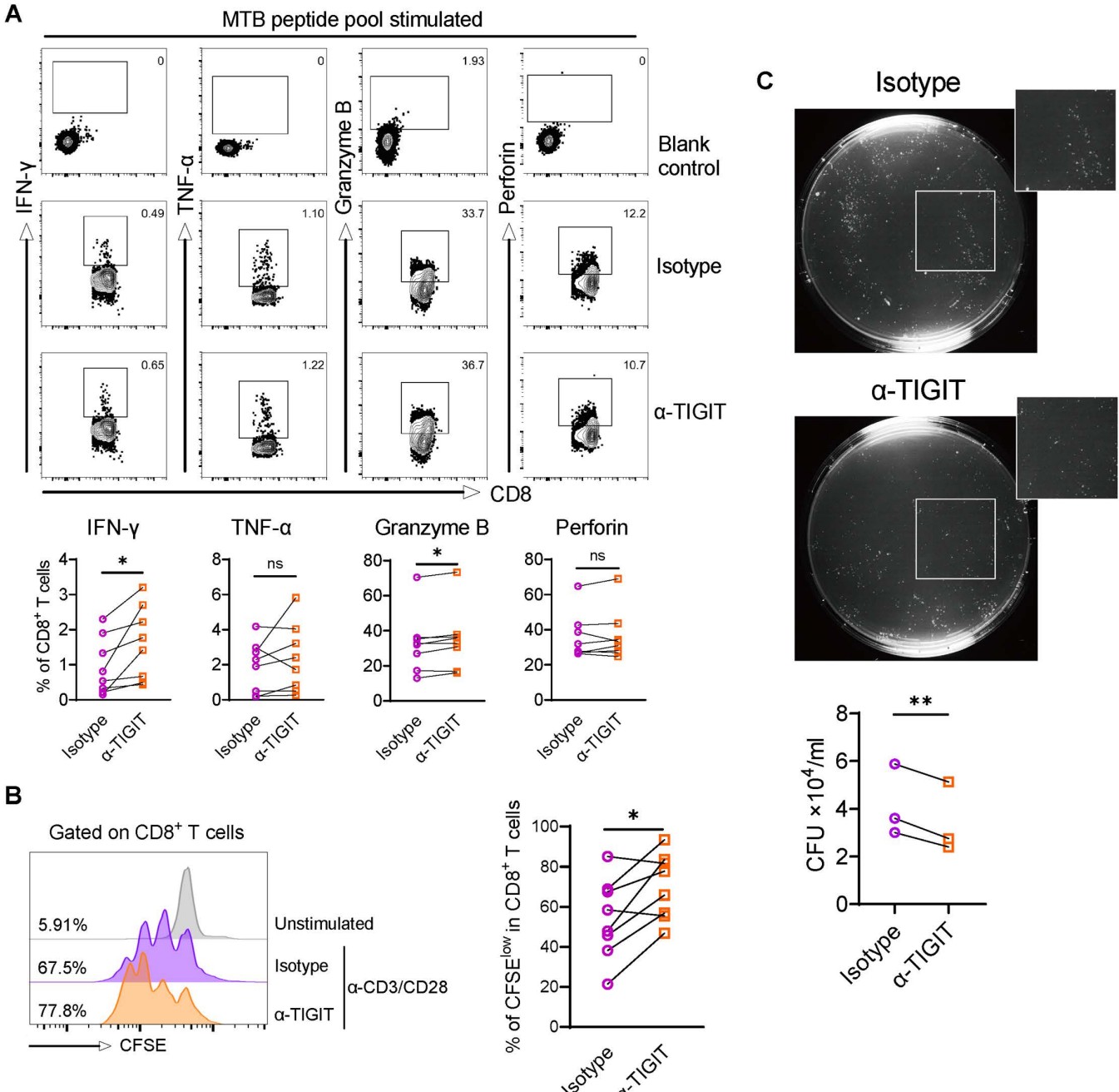

**Fig 4. Blocking TIGIT promotes clearance of mycobacteria by CD8⁺ T cells *in vitro*.** PBMCs from patients with ATB were blocked *in vitro* with anti-TIGIT or isotype control antibody, and then assessed for the production of effector molecules, proliferation, and ability to kill intracellular BCG in infected homologous peripheral blood MDMs. (A) Representative flow contour plots and summarized frequencies of IFN-γ, TNF-α, granzyme B, and perforin expression in CD8⁺ T cells from patients with ATB (n = 8) following MTB peptide pool stimulation in the presence of anti-TIGIT or isotype control antibody. (B) Representative histograms and cumulative data of the percentage of CFSE$^{low}$ proliferating CD8⁺ T cells from patients with ATB patients (n = 8) following anti-CD3/CD28 stimulation in the presence of anti-TIGIT or isotype control antibody. (C) Representative lysate culture plates of BCG-infected (multiplicity of infection 10:1) MDMs after 5-day co-culture with sorted homologous CD8⁺ T cells (Effector:Target = 10:1) from patients with ATB (n = 3) in the presence of anti-TIGIT or isotype control antibody. Data are presented as individual values in panels A and B, and mean ± standard deviation in panel C. Statistical significance ($P < 0.05$) was obtained using a paired t test. *, $P < 0.05$; **, $P < 0.01$; *ns*, not statistically significant.

We also performed bulk RNA-sequencing using sorted CD8+ T cell of ATB patients after *in vitro* treatment with TIGIT blocking antibody and/or MTB peptides. Results showed that TIGIT blockade resulted in a total of 293 and 365 DEGs in MTB-peptide stimulated and unstimulated CD8+ T cells, respectively (S5A Fig). Gene set enrichment analysis revealed that, in CD8+ T cells stimulated with MTB peptides, TIGIT blockade could induce expression of genes associated with response to IFN-γ, inflammatory response and apoptosis, and downstream pathway, including NF-κB signaling, IL-2-STAT5 signaling and mTORC1 signaling (S5B Fig). These results indicated possible mechanisms of the regulatory effect of TIGIT on MTB-specific CD8+ T cells.

### TIGIT blockade promotes clearance of MTB in infected mice

As TIGIT blockade enhanced mycobacterial clearance by activating CD8+ T cell *in vitro*, TIGIT blockade may also promote MTB clearance *in vivo*. We first studied TIGIT expression on T cells after mice were infected intranasally with the MTB H37Rv strain (Fig 5A). TIGIT expression was detected on both CD4+ and CD8+ T cells obtained from spleen of infected mice (Fig 5B). The frequency of TIGIT-expressing cells among splenic CD4+ T cells increased from weeks 2–4 post infection (Fig 5B). TIGIT expression on CD8+ T cells showed a similar trend (Fig 5B). T cells from the blood of MTB infected mice also expressed TIGIT, the frequency of which was higher than in splenic T cells, especially in the CD8+ compartment (Fig 5C). The expression of ligand CD155 was also detected on macrophages and monocytes from the spleens, as well as on monocytes from blood at a comparable level in infected mice (S4A-C Fig). A positive correlation was evident between the frequencies of TIGIT+CD4+ cells among splenic and peripheral T cells with viable splenic CFU counts (S4D Fig).

We next studied whether blocking TIGIT promoted the control of MTB infection in mice. Treatment with TIGIT blocking antibody was begun one week after MTB challenge (Fig 5D). This treatment significantly reduced MTB load by $1.45 \pm 0.19$ $log_{10}$CFU per gram in the lungs, while treatment with anti-PD-1 antibody slightly increased MTB CFU counts by $0.41 \pm 0.17$ $log_{10}$CFU per gram in the lungs (Fig 5E). Viable splenic CFU counts from all three groups of infected mice were comparable (Fig 5E). Assessment of transcriptional levels of effector molecules revealed a significantly higher level of *Ifng* expression in the lung homogenates from mice treated with TIGIT blocking antibody (Fig 5F), which might be associated with the reduced MTB load in lungs. Histopathological assessment of lung sections from the three groups of mice did not reveals obvious difference in the severity of alveolar wall thickening, immune cell infiltration, congestion, and hemorrhage (Fig 5G), which indicated that blocking TIGIT did not significantly aggravate tissue inflammation. The collective findings provide evidence that the blockade of TIGIT, but not PD-1, may enhance immunity against MTB infection *in vivo*.

### Discussion

Reactivation of tuberculosis following immune checkpoint blockade is a newly emerged clinical issue [13]. TIGIT is a next-generation immune checkpoint molecule that is potentially valuable in cancer immunotherapy [24]. TIGIT has been recognized as a negative regulator of T cell function, which participates in the pathogenesis of autoimmune diseases, malignancies, and chronic viral and parasitic infections [25,26]. Upregulation of TIGIT expression has been reported in people and non-human primates with ATB and LTBI [27–29]. However, its role in tuberculosis remains unclear. In this study, we observed upregulated TIGIT expression on CD8+ T cells in blood and infected tissue lesions, and a positive correlation between TIGIT expression and disease severity in both ATB patients and MTB infected mice. TIGIT-expressing cells exhibited higher levels of activation and differentiation, and potent effector molecule production, rather than a profile of "exhaustion", and shared transcriptional features with NK-like cytotoxic effector CD8+T cells. TIGIT blockade improved the proliferation and clearance of intracellular infected BCG *in vitro*, and enhanced the control of MTB infection in a mouse model of tuberculosis. Our results support the negative regulatory role of TIGIT in CD8+ T cell mediated immunity against tuberculosis and demonstrate the promoting effect of TIGIT blockade on the immune response to tuberculosis. Overall, the results indicate targeting of TIGIT may be a therapeutic strategy.

 

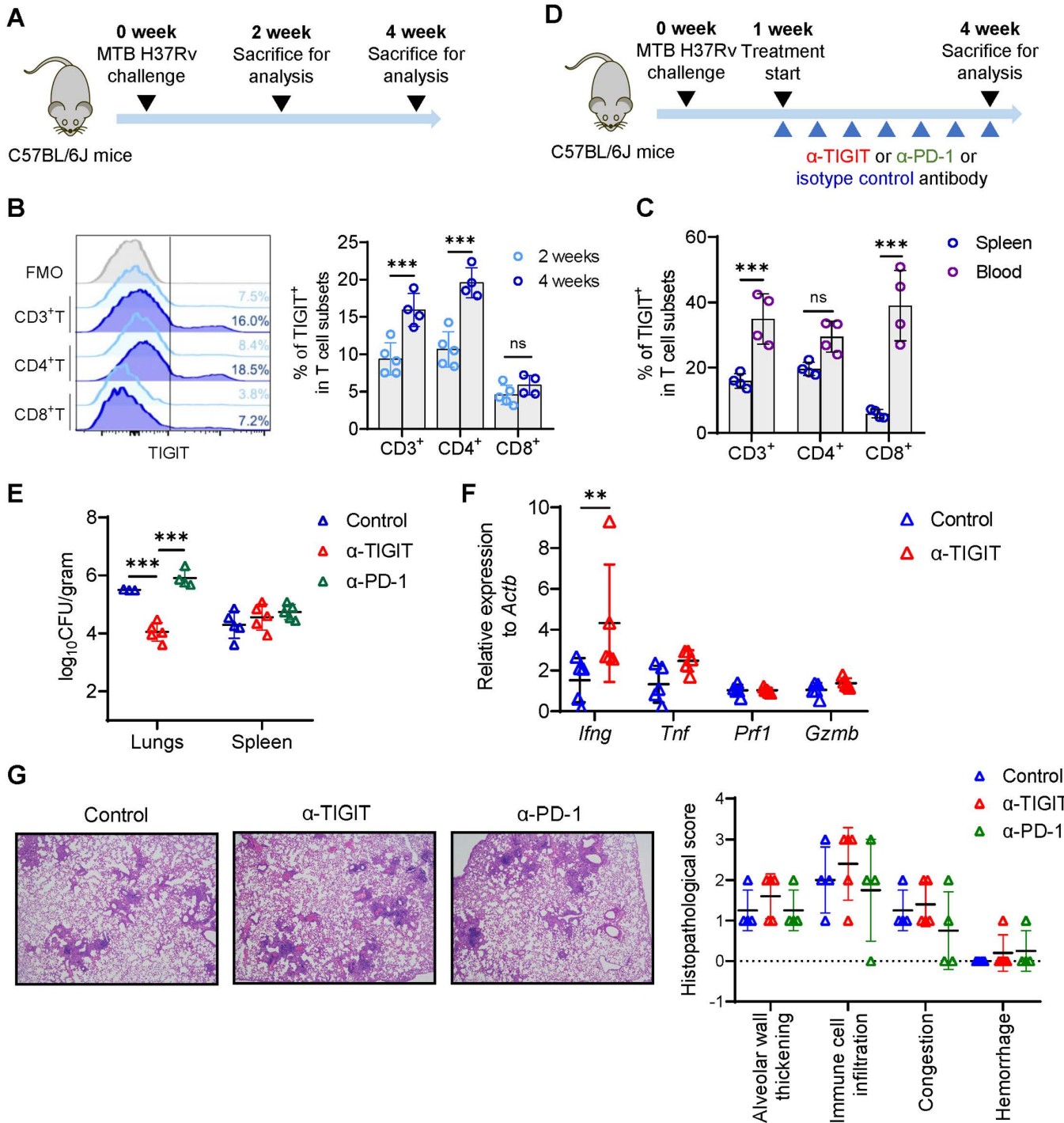

**Fig 5. TIGIT blockade promotes MTB clearance in infected mice.** (A) Experimental schematic. C57BL/6J mice were challenged intranasally with MTB H37Rv strain to establish active tuberculosis infection. TIGIT expression in blood or spleen, and MTB bacterial load in lungs and spleen were assessed by flow cytometry and enumeration of viable cells, respectively. (B) Representative histograms of TIGIT expression and cumulative frequencies of TIGIT+ cells among splenic T cell populations in untreated mice at 2 weeks (n = 5) and 4 weeks (n = 4) after MTB infection. (C) Summarized frequencies of TIGIT+ cells among CD3+, CD4+, and CD8+ T cell populations in spleen (n = 4) and blood (n=4) at 4 weeks post MTB infection. (D) Experimental schematic. *In vivo* antibody blockade was initiated one week following MTB infection via intraperitoneal injection for a total of seven doses. MTB bacterial load in lungs and spleen, and lung histopathological changes assessed by viable cell enumeration counting and hematoxylin and eosin

(H&E) staining, respectively. (E) Summarized viable counts of MTB in lungs and spleens, (F) transcriptional levels of effector molecules in lungs, and (G) representative H&E staining and summarized scores of lesion severity of lung sections from infected mice after 3-week treatment with anti-TIGIT (n = 5), anti-PD-1 (n = 5), or isotype control (n = 5) antibody. Data are presented as mean ± standard deviation in panels B, C, and E-G. Statistical significance ($P < 0.05$) was obtained using an unpaired t test or one-way ANOVA with Bonferroni post-test. ***, $P < 0.001$.

Both the expression of TIGIT on CD8+ T cells and its ligand CD155 on monocytes in the blood from ATB patients were significantly higher as compared to healthy individuals, consistent with previous studies [19,29]. In contrast to these previous studies, we included the LTBI population, which added to the expression profile of TIGIT/CD155 axis in the tuberculosis infection spectrum. The expression levels of TIGIT and CD155 in LTBI individuals were intermediate as compared to ATB and healthy individuals, but were not statistically significant. These results suggest that TIGIT expression increases gradually as MTB infection progresses and was mainly upregulated on the CD8+$T_{em}$ and $T_{emra}$ subsets. Elevation in TIGIT expression on T cells, especially on pathogen-exposed activated differentiated effector subsets, has already been observed in various infections, including acute bacterial infection, and mainly chronic viral and parasitical infections [18,19,30,31]. Data from LCMV infection mouse model also revealed a gradual upregulation of TIGIT expression during viral infection and sustained at a high level on LCMV-specific CD4+ and CD8+ T cells during chronic phase of infection in response to continuous TCR stimulation [32]. Such expression kinetics has been seem in serial of immune checkpoints, including PD-1 and TIM-3 in the context of tuberculosis infection [33,34].

The profile of TIGIT expression in lesions has been described in various malignancies and autoimmune diseases [35], but not in tuberculosis. Here, we report for the first time the expression of TIGIT in lung lesions from individuals with tuberculosis. The finding is consistent with the observation in lung granulomas from MTB infected macaques [28]. Similar to the situation in blood, the upregulation of TIGIT expression in lung lesions has also been seen in a variety of infections [18,19,28,32,36–39]. Our results support a close correlation between TIGIT and the activation of anti-tuberculosis immunity and disease severity of ATB patients. Similar results have been observed in other inhibitory receptors [34,40,41].

We also determined the phenotypic, functional, and transcriptional profiles of the TIGIT+CD8+ T cell population in individuals with ATB. Consistent with previous data [42], most TIGIT-expressing CD8+ T cells in ATB patients were activated effector T cells, while only a small population co-expressed PD-1. High checkpoint receptors co-expression, low effector function, and transcription factor expression patterns are used together as hallmarks of T exhaustion [43]. Of note, the upregulation of a single checkpoint receptor is not sufficient to represent exhaustion, but is more of a sign of CD8+ T cell activation, as immune checkpoints can be transiently expressed on functional effector T cells during activation in acute infection [10]. Therefore, we speculate that TIGIT+CD8+ T cells in ATB patients are not typical exhausted cells, but a group of activated and mature effector cells.

Our functional phenotypic test results support this speculation, as we observed a potent effector production of multiple effector molecules by MTB-specific TIGIT+CD8+ T cells. This speculation is different from what has been found in the context of cancer, where TIGIT is considered a marker for T cell exhaustion and the TIGIT-expressing cell are mostly dysfunctional [44]. Our findings are consistent with a study of TIM-3, which reported greater effector function in producing Th1/Th22 cytokines and cytolytic effector molecules of TIM-3-expressing CD4+ and CD8+ T cells from patients with ATB [34]. Superior functions of TIGIT+CD8+ T cells were also reported in human immunodeficiency virus (HIV) infections [18,45,46]. The transcriptome characteristics of the TIGIT+CD8+ T subpopulation we observed here are similar to the findings of a recent study, which also revealed upregulation in genes associated with antiviral immunity in TIGIT+CD8+ T cells from HIV-infected individuals using transcriptome analysis [46]. Apart from that, we observed higher transcript levels of multiple NK receptor encoding genes in TIGIT+PD-1− CD8+ T cells, which consist with characteristics of a NK cell-like KIR+CD8+ effector T cell subset in cytomegalovirus-infected individuals [47].

We demonstrated that TIGIT blockade could reduce the bacterial load in the lungs of MTB infected mice. This is, to our knowledge, the first data on the effect of *in vivo* TIGIT blockade in a mouse model of tuberculosis. Our results are

consistent with recent descriptions of *in vivo* TIGIT intervention in animal models of other infectious diseases [19,39,48]. TIGIT blockade or knockout can reportedly reduce liver lesions and liver function damage in mice infected with *Alveolar echinococcosis* by enhancing the production of cytokines by T and NK cells [19,39]. In a mouse model of *Schistosoma japonicum* infection, TIGIT knockout also reduced liver fibrosis [48]. Although we observed an elevated *Ifng* transcriptional level in the lungs of TIGIT antibody treated mice, how TIGIT blockade promotes the clearance of infection has not been well explained. Previous studies suggested that the cytokine synthesis of CD8+ T cells does not seem to be restored by *in vivo* TIGIT blockade [19]. Instead, the boosting effect on CD8+ T cell expansion is a possible mechanism, as suggested in a study of simian immunodeficiency virus-infected macaque [18].

In combination with our phenotypical, transcriptional and TIGIT blockade assay results, TIGIT seems to be an activation marker with regulatory effect in the context of active tuberculosis. Similar to study of PD-1, our results support the nuance of immune checkpoint in MTB infection. Study of lung granuloma from MTB infected non-human primates observed a low level of PD-1 expression and well-preserved function of the PD-1+ T cells, suggesting a low level of T cell exhaustion in that microenvironment [49]. PD-1 is now an important target for cancer immunotherapy due to enhancement of anti-tumor immunity upon PD-1 blockade. However, dismissal of the regulatory effect of PD-1 by either gene knock-out or antibody blockade led to dysregulated inflammation, excessive tissue damage, and exacerbation of MTB infection in animal models [50–53]. Besides, there is a growing number of cases with tuberculosis reactivation following PD-1 blockade treatment [13,54]. These observations supported a protective role of the immune regulatory role of PD-1, and suggested a delicate balance between immune suppression and anti-tuberculosis immune response. Therefore, further investigation is required to elucidate the role of TIGIT in CD8+ T-cell immune response against MTB infection.

This study has some limitations. First, there are no data concerning changes in TIGIT expression levels during anti-tuberculosis treatment. Follow-up of the ATB patients should be performed to explore the value of TIGIT as an indicator of treatment efficacy and prognosis of tuberculosis. Second, due to the limitations of experimental conditions and the accessibility of experimental samples, we were unable to explore the functional characteristics of TIGIT+CD8+ T cells in tuberculosis lesions. Moreover, the tuberculosis-infected mouse model used in this study cannot fully simulate the disease and immune response characteristics of human chronic tuberculosis infection. Future studies on the *in vivo* blocking effect of the TIGIT/CD155 axis in primate models of chronic tuberculosis infection will provide more reliable evidence.

In conclusion, the current study reveals the upregulation of TIGIT expression on CD8+ T cells and its correlation with disease severity in ATB. TIGIT-expressing CD8+ T cells from patients with ATB exhibited a polarized activated effector phenotype, instead of features of exhaustion. TIGIT blockade could promote anti-tuberculosis immunity *in vitro* and clearance of MTB infection *in vivo*. Our findings advance the knowledge of the regulatory role of TIGIT in the T cell-mediated immune response to tuberculosis, and have implications for immune intervention in tuberculosis.

## Materials and methods

### Ethics statement

This study was approved by the institutional ethics review board of Huashan Hospital, Fudan University (approval number: HIRB2017–161, HIRB2022–1001). The study was performed in accordance with the guidelines of the Declaration of Helsinki and relevant regulations. Verbal informed consent was obtained from all participants. The animal studies adhered to the ARRIVE reporting guidelines and received approval from the Animal Care and Use Committee of the Shanghai Public Health Clinical Center (approval number: 2018-JS027).

### Study subjects and human sample collection

All participants were recruited from Wuxi Fifth People's Hospital (Jiangsu, China) and Huashan Hospital (Shanghai, China) between 2019 and 2024. Patients with a pathological, histological proven, or clinical diagnosis of ATB that prior

to anti-tuberculosis treatment or who were treated for less than two weeks were included. The LTBI and healthy control groups comprised close contacts of the ATB patients and healthcare workers who underwent screening for tuberculosis. Individuals with LTBI were positive in the interferon-gamma release assay, but showed no clinical evidence of ATB. The healthy control group comprised individuals which were negative in the interferon-gamma release assay without clinical evidence of tuberculosis. Individuals with proven diagnosis of other pulmonary infections who underwent lung lobectomy were also included. Demographic and clinical features of participants were summarized in S1 Table. Peripheral blood samples from individuals in the ATB, LTBI, and healthy control groups were collected. In the immunochemical staining experiment, lung lesions tissue sections collected from individuals with ATB or other pulmonary infection, including two cases with pulmonary cryptococcosis and one case with actinomyces pulmonary infection, were used. These patients were enrolled based on chest CT manifestation of solitary nodules in unilateral or bilateral lobes and chronic granulomatous lesions in lung surgical excision biopsy.

## Animal studies

Six-week-old female C57BL/6J mice purchased from Charles River Laboratories (Beijing, China) were fed in laboratory animal facilities with free access to food and water, using controlled room temperature (22°C±1°C) and humidity (65%±5%), with a standard 12h light-dark cycle. To establish MTB infection, all mice were intranasally infected with 200 CFU of MTB H37Rv strain, using a 40 µL dilution of a log-phase culture. *In vivo* antibody treatment was initiated one week after infection. Mice from the experimental group were treated with a PBS-diluted antibody solution containing 200µg anti-TIGIT (BE0274, Bio X Cell, USA) or anti-PD-1 antibody (BE0146, Bio X Cell, USA) per dose via intraperitoneal injection twice a week for a total of seven doses. Mice from the control group were treated with a same dosage of isotype control antibody (BE0083, Bio X Cell, USA). All treated mice were sacrificed three days after treatment ended. The lungs and spleens were aseptically removed. Organ homogenates were serially diluted and plated in duplicate on selective Middlebrook 7H10 agar (BD, USA) for bacterial quantification. Colonies were counted after three to five weeks. Aliquots of lung homogenate were preserved in TRIzol reagent (Invitrogen, USA) and stored at −80°C for subsequent quantitative PCR analysis. Lung tissue sections were stained with hematoxylin and eosin for histopathological examination. The severity of lung lesions was quantified based on the score of alveolar wall thickening, immune cell infiltration, congestion, and hemorrhage (score=0, none; score=1, slight; score=2, mild; score=3, intermediate; and score=4, severe). Each sample was evaluated in a blinded manner by two pathologists, and conflicting cases were reanalyzed by a third pathologist. A group of untreated mice were sacrificed at two or four weeks after infection, and cells were isolated from blood and spleens for flow cytometry analysis.

## Cell isolation

Human peripheral blood mononuclear cells (PBMCs) were isolated by density gradient centrifugation using Lymphocyte-H (Cedarlane, Canada). Splenocytes and peripheral blood cells from mice were isolated using red blood cell lysis buffer (Biosharp, China). Isolated cells were preserved in Cellbanker2 (ZENOAQ, Japan) at −80°C until analysis.

## Antibodies

Information of the antibodies used in this study is listed in S2 Table.

## Flow cytometry analysis

Cryopreserved cells were thawed and allowed to rest in RPMI 1640 (Corning, USA) supplemented with 10% fetal bovine serum (Gibco, USA) for 30 min. For phenotypic assessment, PBMCs were stained with monoclonal antibodies (mAbs) against surface markers, including CD3, CD8, TIGIT, CCR7, CD45RA, CD25, CD69, PD-1, and TIM-3.

To determine the production of effector molecules, PBMCs were plated at a density of $1 \times 10^6$ cells/well in a 96-well plate and stimulated with 11 ng/µL MTB peptide pool (Fosun Long March, Shanghai, China) as previously described [55], or culture medium. The amino acid sequence of synthetic overlapping peptides of ESAT-6, CFP-10 and RD2 proteins used in this study is listed in S3 Table. After 2 h of stimulation, Golgiplug (BD, USA) was added. After another 14 h of stimulation, cells were collected and stained with mAbs against CD3, CD8, and TIGIT. For intracellular molecule assessment, the cells were permeabilized with Foxp3 Transcription Factor Staining Buffer Set (Invitrogen, USA) following the manufacturor's instructions, and then stained with mAbs against IFN-γ, TNF-α, granzyme B, and perforin.

For analysis of transcription factors, PBMCs were stained with mAbs against surface markers, permeabilized, and then stained intracellularly with mAbs against T-bet and eomesodermin.

For analysis of TIGIT expression in MTB infected mice, splenocytes and peripheral blood cells were first stained using the Zombie NIR Fixable Viability Kit (Biolegend, USA), blocked with TruStain FcX (Biolegend), and then stained with mAbs against CD3, CD4, CD8a, TIGIT, CD45.2, and CD11b. Stained cells were acquired using the Cytoflex S (Beckman Coulter, USA) or Cytek Aurora (Cytek, USA) system, and analyzed using FlowJo v10.7 software (BD, USA).

## Immunohistochemistry staining

Human lung lesions tissue sections were embedded in paraffin. The sections were incubated with anti-human TIGIT primary antibodies (CST, USA, 1:500) at 4°C overnight, followed by incubation with a horseradish peroxidase-conjugated goat anti-rabbit secondary antibody (Abcam, USA, 1:5000) at room temperature for 50 min. The reactive sections were then visualized using 3,3′-diaminobenzidine (DAKO, Denmark, 1:50) and the sections counterstained with hematoxylin. A histochemistry score was used to semi-quantify TIGIT staining for each tissue sample. The score was calculated by multiplying the estimated proportion of staining in lesion lymphocyte aggerations that were positively stained. The intensity score (score = 0, none; score = 1, weak; score = 2, intermediate; and score = 3, strong) was determined using Aipathwell (Servicebio, China). The proportion and intensity scores were multiplied to obtain a total score, which could range from 0 to 300.

## *In vitro* TIGIT blockade

To assess the effect of TIGIT blockade on the production of effecter molecules, PBMCs were plated at a density of $1 \times 10^6$ cells/well in a 96-well plate, and cultured in the presence of 10 µg/mL anti-TIGIT or isotype control antibody for 12 h. The cells were then stimulated with 11 ng/µL MTB peptide pool for 16 h, collected, and stained with mAbs against CD3, CD8, IFN-γ, TNF-α, granzyme B, and perforin, as described above.

## CFSE proliferation assay

PBMCs were stained with 2.5 mM CellTrace CFSE (Invitrogen, USA) according to the manufacturer's protocol. The CFSE-labeled cells were plated at a density of $1 \times 10^6$ cells/well in a 96-well plate pre-coated with 10 µg/mL anti-CD3 antibody, while unstimulated controls were plated in uncoated wells. The cells were cultured for 5 days in the presence of 5 µg/mL anti-CD28 antibody, and 10 µg/mL anti-TIGIT or isotype control antibody. The cultured cells were collected, stained with mAbs against CD4 and CD8, and analyzed using flow cytometry. CFSE[low] cells were assessed to determine the effect of *in vitro* TIGIT blockade on CD8[+] T cell proliferation.

## Cell sorting

For transcriptional analysis, four CD8[+] T cell subsets (TIGIT[+]PD-1[+], TIGIT[+]PD-1[−], TIGIT[−]PD-1[+], and TIGIT[−]PD-1[−]) and CD8[+] T cells cultured in the presence of 11 ng/µL MTB peptide pool plus 10 µg/mL TIGIT blocking antibody, or MTB peptide pool or TIGIT blocking antibody, or culture medium alone for 24h were sorted using PBMCs from four and three

ATB patients, respectively. CD8$^+$ T cells were first isolated from PBMCs using a magnetic negative isolation kit (Miltenyi, Germany) and then stained with allophycocyanin-conjugated anti-TIGIT and BV421-conjugated mAbs against PD-1. Fluorescence-activated cell sorting was performed to enrich the four CD8$^+$ T cell populations using the MoFlo Astrios (Beckman Coulter, USA). The resulting cells were subjected to RNA extraction for transcriptome analysis.

For assessment of the intracellular bacterial killing of CD8$^+$ T cells, CD8$^+$ T cells and CD14$^+$ monocytes were isolated from PBMCs of three patients with ATB by magnetic-activated cell sorting with the use of anti-human CD8 and anti-human CD14 microbeads (Miltenyi, Germany).

### Intracellular BCG killing assay

Peripheral blood MDMs were derived as previously described [56]. CD14$^+$ monocytes sorted from blood of ATB patients were plated at a density of $2 \times 10^4$ cells/well in a 96-well plate, cultured in RPMI 1640 with 10% fetal bovine serum for 48 h to allow macrophage adherence, and washed with PBS. The resulting MDMs were infected with log-phase BCG culture by 3-h phagocytosis at a multiplicity of infection of 10. After infection, MDMs were washed three times with PBS before replacing with the RPMI culture media. Autologous CD8$^+$ T cells sorted from the ATB patients were added to well at an effector to target ratio of 10, in the presence of 10 µg/mL anti-TIGIT or isotype control antibody. After a 5-day co-culture, cells were washed three times with PBS and lysing in sterile-filtered distilled water with 0.025% sodium dodecyl sulfate (Sigma-Aldrich, Germany) for 10 min at 37°C to determine the viable counts per well as CFU. Ten-fold serial dilution of the resulting cell lysates prepared with PBS were plated on 7H10 plates supplemented with 0.5% glycerol and 10% oleic acid-albumin-dextrose-catalase, and incubated for 21–28 days at 37°C.

### Transcriptome analysis

RNA extracted from four sorted CD8$^+$ T cell subsets (TIGIT$^+$PD-1$^+$, TIGIT$^+$PD-1$^-$, TIGIT$^-$PD-1$^+$, and TIGIT$^-$PD-1$^-$) or CD8$^+$ T cell sorted from PBMC treated with MTB peptides and/or TIGIT blocking antibody from patients with ATB was converted to cDNA libraries using the TruSeq RNA sample prep Kit (Illumina, USA) according to the manufacturer's instructions. The libraries were then sequenced using the HiSeq X Ten system (Illumina). The quality of the resulting data was assessed using FastQC software. The raw sequencing reads were pre-processed and the filtered clean reads were aligned to the human reference genome using TopHat2. Uniquely mapped reads were then assigned to each annotated gene using featureCounts in R. Differential expression analyses were performed using DESeq2. Annotated sequences with absolute log$_2$-transformed fold changes > 1 and adjusted $P < 0.05$ were considered as differentially expressed genes. A heatmap for the visualization of gene expression and a normalized enrichment score were generated using Morpheus (https://software.broadinstitute.org/morpheus). The Gene Ontology and Kyoto Encyclopedia of Genes and Genomes enrichment analysis of the differentially expressed genes, and gene set enrichment analysis were performed using R studio, SangerBox (http://www.sangerbox.com/) and visualized using the OmicStudio tools (https://www.omicstudio.cn/tool).

### Quantitative PCR analysis

Total RNA was extracted from aliquots of lung homogenate of treated mice preserved in TRIzol reagent. The cDNA was synthesized using PrimeScript RT Master Mix (Takara, Japan). Quantitative real-time PCR was performed using the TB Green Premix Ex Taq II kit (Takara). The relative expression of *Ifng* (F 5'-ATGAACGCTACACACTGCATC-3', R 5'-CCATCCTTTTGCCAGTTCCTC-3'), *Tnf* (F 5'-GCTACGACGTGGGCTACAG -3', R 5'-CCCTCACACTCA GATCATCTTCT-3'), *Prf1* (F 5'- AGCACAAGTTCGTGCCAGG-3', R 5'-GCGTCTCTCATTAGGGAGTTTTT-3'), and *Gzmb* (F 5'-CCACTCTCGACCCTACATGG-3', R 5'-GGCCCCCAAAGTGACATTTATT-3') in the lung homogenate from TIGIT blocking antibody treated mice was calculated using the ΔΔC$_t$ method, with *Actb* (F 5'- GGCTGTATTCCCCTCCATCG-3',

R 5'-CCAGTTGGTAACAATGCCATGT-3') serving as an internal control and lung homogenate from isotype control antibody treated mice serving as the control group.

## Statistical analyses

Data are presented as the mean±standard deviation or median with IQR. GraphPad Prism 8.0 Software (San Diego, CA, USA) was used for calculations, statistical analyses, and graphic generations. The CFU counts ($x$) underwent a logarithmic transformation as $\log_{10}(x+1)$. Statistical analyses comparing two parameters were performed using the unpaired t (Mann-Whitney U) test or the paired t (Wilcoxon signed rank) test. Statistics for multiparameter analyses were determined by one-way analysis of variance (ANOVA) with Bonferroni post-test (Kruskal-Wallis with Dunn post-test). $P<0.05$ was considered significant.

## Supporting information

**S1 Data. Annotated differentially expressed genes in TIGIT+PD-1+CD8+ T cell subset compared to the TIGIT−PD-1+ subset.**
(XLSX)

**S2 Data. Annotated differentially expressed genes in TIGIT+PD-1−CD8+ T cell subset compared to the TIGIT−PD-1− subset.**
(XLSX)

**S1 Table. Demographic and clinical features of enrolled participants.**
(DOCX)

**S2 Table. Antibodies and usage.**
(XLSX)

**S3 Table. Amino acid sequence of synthetic overlapping peptides of ESAT-6, CFP-10 and RD2 proteins.**
(DOCX)

**S1 Fig. Frequencies of TIGIT-expressing cells on blood CD4+ T cells among patients with ATB (n=37) with different symptoms and the results of interferon-gamma release assay (IGRA), MTB test, and chest computed tomography scans.** Symptomatic refers to cough over 2weeks, chest pain, fatigue, weight loss, fever, night sweats, dyspnea, and/or hemoptysis. Data are presented as median with interquartile range. Statistical significance ($P<0.05$) was obtained using a Mann-Whitney U test. *ns*, not statistically significant.
(TIF)

**S2 Fig. Expression of TIGIT and other inhibitory receptors on MTB-responsive CD8+ T cells.** (A) Expression of TIGIT, PD-1 and TIM-3 on IFN-γ- and/or granzyme B-producing CD8+ T cells, and (B) PD-1 and TIM-3 expression on IFN-γ+ and IFN-γ− TIGIT+ CD8+ T-cell subsets from individuals with ATB (n=8) upon MTB peptide pool stimulation. (C) Comparison of TIGIT, PD-1 and TIM-3 expression on CD8+ T cells from individuals with ATB (n=8) with or without MTB peptide pool stimulation. Data are presented as mean±standard deviation in panel A, and as individual values in panel B and C. Statistical significance ($P<0.05$) was obtained using a one-way ANOVA with Bonferroni post-test or paired t test. *, $P<0.05$; **, $P<0.01$; ***, $P<0.001$; *ns*, not statistically significant.
(TIF)

**S3 Fig. Representative histograms and cumulative level of Ki-67 expression among TIGIT+ and TIGIT−CD8+ T cell populations from blood in individuals with ATB (n=8).** Data are presented as individual values. Statistical significance ($P<0.05$) was obtained using a Wilcoxon signed rank test. *, $P<0.05$.
(TIF)

**S4 Fig. CD155 expression and correlation of TIGIT expression and bacterial burden in MTB infected mice.** (A) Gating strategy and (B) representative histograms and cumulative frequencies of CD155+ cells among spleen macrophages and monocytes in isotype control antibody treated mice during MTB infection for 2 weeks (n = 5) and 4 weeks (n = 4). (C) Cumulative frequencies of CD155+ cells among monocytes in spleen (n = 5) and blood (n = 4) samples from MTB infected mice at 4 weeks. (D) Scatter plots of the frequencies of TIGIT+ CD4+ cells in spleen (n = 17) and peripheral (n = 8) T cells against spleen MTB viable counts in infected mice. Data are presented as mean ± standard deviation in panels B and C, and individual values in panel D. Statistical significance ($P < 0.05$) was obtained using a Wilcoxon signed rank test or Spearman correlation analysis. ***, $P < 0.001$; *ns*, not statistically significant.
(TIF)

**S5 Fig. Bulk RNA-seq and gene expression analysis were performed using CD8+ T cell sorted from PBMC treated with MTB peptides and/or TIGIT blocking antibody from patients with ATB (n = 3).** (A) Volcano plots of differentially expressed gene (DEGs) related to TIGIT blockade in CD8+ T cells with or without MTB peptide stimulation. (B) Gene sets enriched in CD8+ T cells in response to *in vitro* TIGIT blockade. MA, MTB peptide stimulation plus *in vitro* TIGIT blocking antibody treatment; M, MTB peptide stimulation alone; A, *in vitro* TIGIT blocking antibody treatment alone; NON, culture medium alone.
(TIF)

## Author contributions

**Conceptualization:** Qingluan Yang, Qiaoling Ruan.

**Data curation:** Jingyu Zhou, Ning Jiang.

**Formal analysis:** Jingyu Zhou.

**Funding acquisition:** Qingluan Yang, Yuanyuan Liu, Qiaoling Ruan.

**Investigation:** Jingyu Zhou, Qingluan Yang, Haoxin Xu, Huaxin Chen, Mengqing Qian, Xing Lin, Yixuan Yang, Feiran Zhou, Yuzhen Xu, Qianqian Liu.

**Methodology:** Ning Jiang, Qianqian Liu, Yuanyuan Liu, Yan Gao.

**Project administration:** Qiaoling Ruan.

**Resources:** Huaxin Chen, Qinfang Ou, Yan Gao, Wenhong Zhang, Lingyun Shao.

**Software:** Ning Jiang.

**Supervision:** Wenhong Zhang, Lingyun Shao.

**Validation:** Haoxin Xu, Mengqing Qian, Yuzhen Xu.

**Visualization:** Jingyu Zhou.

**Writing – original draft:** Jingyu Zhou.

**Writing – review & editing:** Qingluan Yang, Haoxin Xu, Ning Jiang, Lingyun Shao, Qiaoling Ruan.

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
