## [Decision Letter · Decision Letter 0]

PPATHOGENS-D-24-02617

TIGIT blockade improves anti-Mycobacterium tuberculosis immunity

PLOS Pathogens

Dear Dr. Ruan,

Thank you for submitting your manuscript to PLOS Pathogens. After careful consideration, we feel that it has merit but does not fully meet PLOS Pathogens's publication criteria as it currently stands. Therefore, we invite you to submit a revised version of the manuscript that addresses the points raised during the review process. Both reviewers commented on the novelty of the work, and its potential significance.  Of particular concern, were the reviewers comments around a broader consideration of check-point inhibitors in the context of infection with Mtb, demonstration that the responses seen reflect antigen-driven responses to Mtb, and well as the concerns around some of the experimental details of the report.

Please submit your revised manuscript within 60 days Mar 29 2025 11:59PM. If you will need more time than this to complete your revisions, please reply to this message or contact the journal office at plospathogens@plos.org. Please include the following items when submitting your revised manuscript:

We look forward to receiving your revised manuscript.

Kind regards,

David M. Lewinsohn

Academic Editor

PLOS Pathogens

Michael Wessels

Section Editor

PLOS Pathogens

 Sumita Bhaduri-McIntosh

Editor-in-Chief

PLOS Pathogens

orcid.org/0000-0003-2946-9497

Michael Malim

Editor-in-Chief

PLOS Pathogens

orcid.org/0000-0002-7699-2064

**Journal Requirements:**

1) Thank you for including an Ethics Statement for your study. Please include:

i) A statement that formal consent was obtained (must state whether verbal/written).

3) Please upload a copy of Figure S4 (A-D) which you refer to in your text on page 11. Or, if the figure is no longer to be included as part of the submission please remove all references to it within the text.

4) We have noticed that you have uploaded Supporting Information files, but you have not included a complete list of legends. Please add a full list of legends for your (Original data) file after the references list.

Potential Copyright Issues:

i) Figure 5A. Please confirm whether you drew the images / clip-art within the figure panels by hand. If you did not draw the images, please provide (a) a link to the source of the images or icons and their license / terms of use; or (b) written permission from the copyright holder to publish the images or icons under our CC BY 4.0 license. Alternatively, you may replace the images with open source alternatives. See these open source resources you may use to replace images / clip-art:

7) Please ensure that the funders and grant numbers match between the Financial Disclosure field and the Funding Information tab in your submission form. Note that the funders must be provided in the same order in both places as well. These grants "81801975, 22YF1404900,GWVI-11.1-07, and HS2021SHZX001" are missing from the Funding Information tab while these grants "82302533 and 82101852" are missing from the Financial Disclosure field.

Please indicate by return email the full and correct funding information for your study and confirm the order in which funding contributions should appear. Please be sure to indicate whether the funders played any role in the study design, data collection and analysis, decision to publish, or preparation of the manuscript.

**Reviewers' Comments:**

Reviewer's Responses to Questions

**Part I - Summary**

Reviewer #1: In this manuscript, Zhou and colleagues investigated the role of TIGIT in tuberculosis. The authors found higher TIGIT expression on circulating CD8 T cells in blood from patients with ATB than compared to patients with LTBI and healthy controls. The authors conclude that circulating TIGIT+ CD8+ T cells resemble an activated phenotype rather than an exhausted phenotype. The authors also assess the use of anti-TIGIT antibodies in vitro and in vivo. Incubation with anti-TIGIT antibody in vitro resulted in increased IFNg and granzyme B production, albeit small, in CD8 T cells. Treatment of H37Rv-infected mice anti-TIGIT antibody initiated one week after infection resulted in a reduction in lung CFU, but no reduction of spleen CFU. This study is the first to characterize TIGIT in blood and lung lesions from patients with TB. While these finding are novel and the in vivo anti-TIGIT antibody experiments are interesting, the discussion regarding the nuance of TIGIT over the course of infection (i.e., chronic and acute infection) as well as providing context of the literature of other immune checkpoint receptors (e.g., PD-1) in TB is under-developed.

Reviewer #2: Summary: The authors characterize TIGIT expression on CD4 and CD8 T cells in patients with active and latent tuberculosis to assess their contribution to TB control. They use a series of assays to evaluate immune cell phenotype, effector function, proliferative capacity and transcriptional profile to interrogate T cell subsets with respect to TIGIT expression. Additionally, they evaluated the correlation between TIGIT expression and clinical severity of TB in patients. They present convincing data that TIGIT blockade improves proliferation and cytokine/cytoxic function in CD8+ T cells in samples from patients with active TB. Although, TIGIT has been described as an exhaustion marker for T cells in the literature, the authors present data that TIGIT expression is a marker of activation in TB rather than representing a state of exhaustion. The authors further support their human data with in vivo studies in mice demonstrating that TIGIT blockade results in reduced Mtb CFU but does not induce more immunopathology in the lungs. Their study contributes to our understanding of the role of immune checkpoints in tuberculosis immunity and immunopathology. Overall this is an interesting study, well written and clear scientific reasoning. However, several key issues should be addressed:

**Part II – Major Issues: Key Experiments Required for Acceptance**

Reviewer #1: 1) There is limited acknowledgment in the introduction and discussion to the nuance of immune checkpoint receptors as it relates to immune activation, exhaustion and tuberculosis. For example, immune checkpoint receptors like PD-1 are upregulated during acute infection during initial immune activation. In contrast, in chronic infections, co-expression of multiple immune checkpoint receptors is used to identify exhausted cells. It would improve context to provide discussion of the literature of TIGIT over the course of infection and in contrast to other immune checkpoint receptors.

In addition, it is unclear in the manuscript whether TIGIT is framed as an activation marker or an exhaustion marker. At times in the manuscript, TIGIT is compared to PD-1, which would imply its relation to immune exhaustion, and yet there is limited discussion regarding the body of work investigating PD-1 in TB (PMID: 33452107, 20624978, 32091388, 30651320, 29891540).

2) In Fig 1D, TIGIT expression is compared between symptomatic and asymptomatic patients with TB. Active TB is typically defined as exhibiting symptoms, so it is not possible to be asymptomatic with active TB. In addition, symptoms are not defined for “symptomatic” patients. In Fig 1E, “other pulmonary infection” is not defined and, thus, the rationale for making such a comparison is not clear.

3) There is limited rationale provided in the introduction as to why the authors chose to focus their study on CD8 T cells.

Reviewer #2: 1. The human data evaluates TIGIT on bulk CD4 and CD8 T cell responses but does not show data on the role of TIGIT in antigen specific CD8+ T cell function, which is important for interpreting their ability to have a protective impact on TB immunity. Additionally, among effector molecule-expressing T cells, what frequency are TIGIT+ or TIGIT-? This question is slightly different than what the authors an answer in Figure 2. This will help to answer what is the role of TIGIT expressing T cells in TB since it is well established that antigen-specific T cells are important for TB control. Additionally, how does peptide stimulation alter the TIGIT, TIM-3, PD-1 profiles? What are the TIGIT, TIM-3, PD-1 profiles of antigen specific cells (Figure 2).

2. Line 178/ Figure 4 (line 692): The title of section/figure is misleading because the co-culture was performed with BCG and not Mtb. Although there is some overlap between BCG and Mtb, BCG lacks many of the pathogenic factors that contribute the Mtb replication and immune suppression. Therefore, the authors need to use Mtb as this may generate different data than what is presented. Additionally, the CFU differences between isotype and TIGIT blockade are very modest (Figure 4C). More image contrast is needed in the representative images (Figure 4C) to observe/assess the visual differences.

3. Figure 5: similar to the human data, there is no assessment of antigen specific T cells responses that could be evaluated with Mtb peptide stimulation in the murine cells. Additionally, there is no effector function readout (IFN-gamma, TNFa, granzyme, etc.). Although there is data on T cells in the blood and spleen, T cell lung responses are integral in the control of Mtb and are not provided. Functional assays should be performed on infected mouse lung homogenates to elucidate why TIGIT blockade results in a CFU difference in the lungs but not the spleen.

4. Figure 5D/ Supplementary Table 2/Animal studies methods: the anti-mouse TIGIT blocking antibody details are not provided (clone, manufacturer), also in vivo dosing and timing of 7 administrations should be explicitly stated.

5. Figures 5B and 5C: The axis should be labeled “% of TIGIT+ in T cell subsets” not “in CD8+ T cell subsets”, since this is a comparison between bulk CD3+ T cells and CD4+ and CD8+ T subsets.

6. Clarify content of the MTB peptide pool (peptide sequence/ protein origin). Are these overlapping peptides? To which antigens do these peptides correspond? This information is needed to inform readers about expected stimulation patterns especially among TB researchers.

7. Figure 3: Why was bulk RNAseq performed based on co-expression of TIGIT and PD-1 in ATB patients only? Flow data suggested that PD-1 had minor impact on TIGIT+ cells. RNAseq would have been more informative if populations from Mtb-peptide stimulated, and TIGIT blockade groups were compared to non-stimulated conditions. Alternatively, sequencing could have been performed between ATB, LTBI, and HC TIGIT+ CD8+ T cell populations. This would have been more insightful to make claims regarding TIGIT role in the context of Mtb.

**Part III – Minor Issues: Editorial and Data Presentation Modifications**

Reviewer #1: 4) Mycobacterium tuberculosis is a Gram Positive Bacteria, not Gram Negative. The manuscript Section/Category should be changed to reflect this.

5) Refs 24 & 25 appear to cite dissertation work and not published or accepted manuscripts. Suggest removing these reference and related text.

6) Line 80, insert “non-human” before primate.

7) Line 101, awkward wording: suggest changing “distinct MTB infection statuses” to “either ATB or LTBI”.

8) Line 126, insert “active” between “during” and “tuberculosis”.

9) Lines 188 & 449, change “homologous” to “autologous”.

10) Line 189, include the numeric difference in CFU between TIGIT blockade and isotype in the text.

11) Line 197, change “MTB clearance” to “mycobacterial clearance”. Experimentally, TIGIT blockade reduced BCG CFU, which is Mycobacterium bovis not MTB.

12) Lines 200-201, change “the spleens” to “spleen”.

13) Line 202, change “CD4+ spleen T cells” to “splenic CD4+ T cells”.

14) Lines 209-210, change “with the MTB viable counts in infected mouse spleens” to “with viable splenic CFU”.

15) Line 213, include the numeric difference in CFU between TIGIT blockade and isotype in the text.

16) Line 232, change “organ” to “tissue”

17) Line 242, typo – change to “Both”

18) Line 269, typo – change to “speculation”

Reviewer #2: n/a

PLOS authors have the option to publish the peer review history of their article (what does this mean? ). If published, this will include your full peer review and any attached files.

**Do you want your identity to be public for this peer review?** For information about this choice, including consent withdrawal, please see our Privacy Policy .

Reviewer #1: No

Reviewer #2: No

**Figure resubmission:**
---

## [Decision Letter · Decision Letter 1]

Dear Dr. Ruan,

We are pleased to inform you that your manuscript 'TIGIT blockade improves anti-Mycobacterium tuberculosis immunity' has been provisionally accepted for publication in PLOS Pathogens.

Best regards,

David M. Lewinsohn

Academic Editor

PLOS Pathogens

Michael Wessels

Section Editor

PLOS Pathogens

Sumita Bhaduri-McIntosh

Editor-in-Chief

PLOS Pathogens

orcid.org/0000-0003-2946-9497

Michael Malim

Editor-in-Chief

PLOS Pathogens

orcid.org/0000-0002-7699-2064

Reviewer Comments (if any, and for reference):

Reviewer's Responses to Questions

**Part I - Summary**

Reviewer #1: (No Response)

Reviewer #2: The authors have address most of the concerns. A few minor Editorial Changes:

Line 205: Clarify the CFU reduction. The language is unclear. Is 0.61 the difference between the groups, and 0.85 is the resulting value after blockade?

Line 349: change to “diagnosed cases” or “cases” instead of “cases diagnosed”

**Part II – Major Issues: Key Experiments Required for Acceptance**

Reviewer #1: 1) …We have revised the corresponding part in the “Introduction” and “Discussion” section to discuss more about the nuance of immune checkpoint receptors and current knowledge about TIGIT (Line 96-98, 279-287). Discussion about current knowledge in the role of PD-1 in tuberculosis and the mentioned references has been added to the corresponding part in the “Discussion” section (Line 341-354).

The authors address the nuance of immune checkpoint receptors in the Introduction and Discussion. This helps to provide context for the reader and these changes make for a stronger manuscript.

2) ..We have revised the corresponding part in the figure legends of Figure 1 and Figure S1…In Figure 1E, individuals with other pulmonary infection comprised two cases with pulmonary cryptococcosis and one case with actinomyces pulmonary infection… We have revised the corresponding part in the “Materials and Methods” section (Line 398-401).

The authors have defined symptomatic patients and patients with other pulmonary infection in the text.

3) …We have also added current understanding and research gaps in the study of CD8+ T-cell immune response in MTB infection to the “introduction” section (Line 81-91).

The authors have provided sufficient rationale for focusing on CD8 T cells.

Reviewer #2: none

**Part III – Minor Issues: Editorial and Data Presentation Modifications**

Reviewer #1: The authors have sufficiently addressed all minor edits.

Reviewer #2: Line 205: Clarify the CFU reduction. The language is unclear. Is 0.61 the difference between the groups, and 0.85 is the resulting value after blockade?

Line 349: change to “diagnosed cases” or “cases” instead of “cases diagnosed”

PLOS authors have the option to publish the peer review history of their article (what does this mean? ). If published, this will include your full peer review and any attached files.

**Do you want your identity to be public for this peer review?** For information about this choice, including consent withdrawal, please see our Privacy Policy .

Reviewer #1: No

Reviewer #2: No

---

## [Editor Report · Acceptance letter]

Dear Dr. Ruan,

We are delighted to inform you that your manuscript, "TIGIT blockade improves anti-Mycobacterium tuberculosis immunity," has been formally accepted for publication in PLOS Pathogens.

Best regards,

Sumita Bhaduri-McIntosh

Editor-in-Chief

PLOS Pathogens

orcid.org/0000-0003-2946-9497

Michael Malim

Editor-in-Chief

PLOS Pathogens

orcid.org/0000-0002-7699-2064